# Mobile Robots Autonomous Exploration with Reinforcement Learning

**Haojie Shi**
Department of Electrical Engineering
The Chinese University of Hong Kong
Shatin, Hong Kong
h.shi@link.cuhk.edu.hk

**Ang Li**
Department of Electrical Engineering
The Chinese University of Hong Kong
Shatin, Hong Kong
psw.liang@link.cuhk.edu.hk

## Abstract

Reinforcement Learning, a computational approach to learning whereby an agent tries to maximize the total amount of reward it receives while interacting with a complex and uncertain environment. And it has quite a few applications in playing game or controlling a robot after exploration and exploitation. Our project is aimed at designing an efficient robot autonomous exploration algorithm based on reinforcement learning. Our work is based on the HouseExpo dataset and we developed reinforcement learning algorithm on it, showing that the robot can efficiently explore uncertain environments with a smart behaviour. Our work is based on Double DQN with proportional prioritization[7], and we improve the algorithm by encoding history observation, adding global explored map and designing external reward via Artificial Potential Field. The video is in video link.
**Key words: Q Learning, Autonomous Exploration, Artificial Potential Field**

## 1  Introduction

An extremely significant part of the SLAM task is to require the robot to traverse the environment autonomously and build a map. Due to the environments is different, the robot's exploration strategies probably be completely different. For example, the exploration of ruins after disasters requires robots to explore areas with high risks and alive targets as much as possible. Besides, some exploration missions require exploration to have high overlap and precision. Therefore, it is difficult to provide a sliver bullet algorithm using traditional planning methods. However, if the robot's motion policy system is based on reinforcement learning, it can use methods like transfer learning to achieve an acceptable result by modifying the reward function or observing retraining.

This article uses the HouseExpo data set [3] for training indoor robot exploration task. The data set is collected from indoor areas such as office or school which contains many rooms and corners. The task requires the robot navigation strategy have relatively high area coverage. The traditional strategy is to select an unknown area nearby through greedy algorithm, and then navigate from the current location to the target location. Such algorithms will cause the robot to ignore the value of the unknown area. Therefore, the aforementioned requirement of cannot be met.

This article proposes a method based on DQN with information gain and artificial potential field assistant reward. We have done several experiments to prove that our proposed method has better results than traditional methods. Moreover, Another contribution of this paper is the discovery that in the exploration task of complex environment, the exploration rate reward is too sparse, which leads to the problem that the training is difficult to converge.

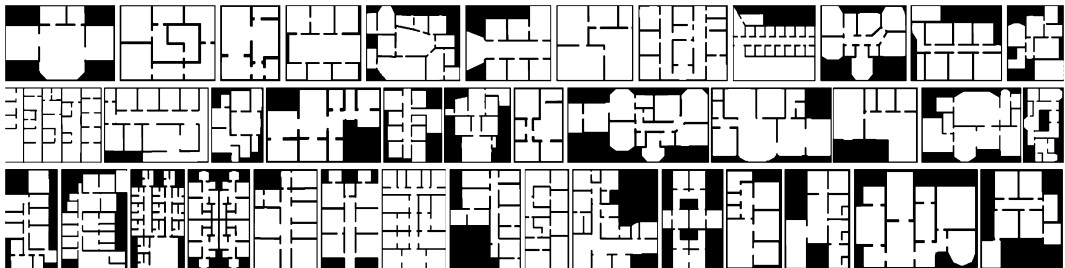

Figure 1: HouseExpo Dataset

## 2   Related Work

In traditional methods, the robot exploration problem can be divided into two parts: optimal target determination and trajectory navigation.

### 2.1   Optimal Target Determination

FrBdkric Bourgault et al.[1] firstly used Shannon information entropy to define the optimal target in robot exploration. After that, Pimentel, Jhielson et al.[5] raised an algorithm based on frontier and mutual information. The algorithm predicts the information entropy for each frontier from unknown area, and then computes the mutual information between the current action sequence and the possible next action. Therefore, the robot navigation policy can explore more unknown areas in a short time. In 2019, Rakesh Shrestha et al.[6] proposed a method called Learned Map Prediction Based Robot Exploration. This paper uses the Learned map to guide reinforcement learning, so that the algorithm can converge in a short period.

### 2.2   Trajectory navigation

For the trajectory planning algorithms, the artificial potential field method proposed by Khatib et al.[4] calculates the gravitational field of obstacles so that the trajectories can reach an approximate global optimum. Since the artificial potential field method has high computational complexity when facing on complex obstacles, Steven M. LaValle et al.[2] proposed the Rapidly-Exploring Random Trees method. The algorithm can quickly plan the trajectory between the current position and the target position through random sampling and collision detection.

Overall, The method based on reinforcement learning used in this article can merge the two parts into a single policy. The algorithms planning the trajectories while obtaining a better target position.

## 3   Problem Definition

The object of our autonomous exploration problem is to find an optimal path $\xi^*$ to explore the whole target area. For grid-map based planning, the path $\xi$ is decomposed into a set of way points $F = \{x_1, ..., x_T\}$ and the goal is thus transformed into finding an optimal point set $F^*$. This optimization can be formalized as follows

$$F^* = \max_{x_{1:T}} \sum_{i=1}^{T} I(x_i|x_{1:i-1}) - \lambda L(x_i, x_{i-1}) \tag{1}$$

where $I(x_i|x_{1:i-1})$ means the information gain in way point $x_I$ given explored points $x_{1:i-1}$, and $L(x_i, x_{i-1})$ means the length between point $x_I$ and $x_{i-1}$. The formulation means that we want to explore as large as possible areas in as few as possible steps.

Transforming this optimization problem into a reinforcement learning task, we first denote $o_t$ as a local map with the size of (64,64) at time $t$, and the action space $a_t$ is discretized as three bins, containing $forward, left$ and $right$. Besides, the reward function is defined as follows

$$R(a_t, s_t) = I(a_t|s_t) \tag{2}$$

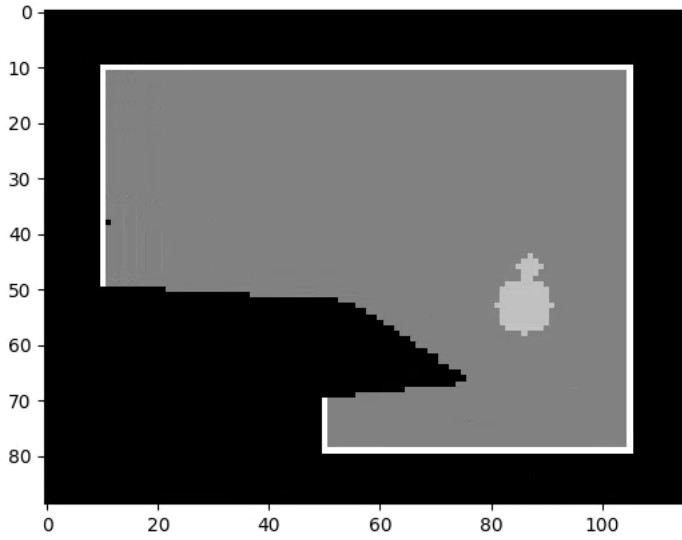

Figure 2: Robot Autonomous Exploration

where the term $I(a_t|s_t)$ is information gain with action $a_t$ given the state $s_t$. And we constraint the maximum steps of one episode as 200, so as to force the agent explore more areas within a given time. Therefore, the Reinforcement learning object now is formalized as follows

$$\pi(a_t|s_t) = \arg\max_{a_t} \sum_{m=t}^{T} \gamma^{m-t} R(a_t, s_t) \tag{3}$$

$$s.t. T <= 200$$

## 4 Proposed Approach

### 4.1 Baseline: Double DQN with proportional prioritization

The algorithm is shown as Alg.1. For reinforcement learning, building the replay memory is a general approach to store millions of transition pairs and is used for stochastic gradient descent. Basically, the transitions are selected uniform randomly for gradient descent. However, picking the transitions of the replay memory in a good order can lead to exponential speed-ups over uniform choice. While prioritizing with TD error is an alternating approach in online RL algorithms, such prioritizing method can suffers from pool estimation of TD error in some circumstances, e.g. when rewards are noisy, resulting in unsampling some worthwhile transitions. In [7], a stochastic sampling method was introduced that interpolates between pure greedy prioritization and uniform random sampling. The intrinsic idea is that the probability of being sampled is monotonic in a transition's priority, while a non-zero probability even for the lowest-priority transition is guaranteed. Concretely, the probability of sampling transition $i$ is defined as

$$P(i) = \frac{p_i^\alpha}{\Sigma_k p_k^\alpha} \tag{4}$$

### 4.2 Global Observation

As illustrated above, our observation space only contain a local map with the size of (64,64) like Figure 2. However, if the robot can only observe the local map in current frame, it's a common case that the robot may not be able to find a good path to leave if it gets stuck in the corner. Adversely, if the robot can receive a sequence of the observation in last several frames, it would be aware of its situation and leave the corner instead of repeating previous action and moving in circles. Therefore, to make the robot aware of past state, the last 4 frames of local map is stacked and reshaped as a

---

**Algorithm 1** Double DQN with proportional prioritization

---

**Input:**
minibatch $k$, step-size $\eta$, replay period $K$ and size $N$, exponents $\alpha$ and $\beta$, budget $T$.
1: Initialize replay memory $\mathcal{H} = \Phi, \Delta = 0, p_i = 1$
2: Observe $S_0$ and choose $A_0 \sim \pi_\theta(S_0)$
3: **for** $t = 1$ **to** $T$ **do**
4:     Observe $S_t, R_t, \gamma_t$
5:     Store transitions $(S_{t-1}, A_{t-1}, R_t, \gamma_t, S_t)$ in $\mathcal{H}$ with maximal priority $p_t = max_{i<t} p_i$
6:     **if** $t \equiv 0 \bmod K$ **then**
7:         **for** j=1 **to** k **do**
8:             Sample transition $j \sim P(j) = p_j^\alpha / \Sigma_i p_i^\alpha$
9:             Compute importance sampling weight $w_j = (N(j))^{-\beta} / max_i w_i$
10:            Compute TD-error $\delta_j = R_j + \gamma_j Q_{target}(S_j, arg\,max_a\, Q(S_j, a)) - Q(S_{j-1}, A_{j-1})$
11:            Update transition priority $p_j \leftarrow |\delta_j|$
12:            Accumulate weight-change $\Delta \leftarrow \Delta + w_j \cdot \delta_j \cdot \bigtriangledown_\theta Q(S_{j-1}, A_{j-1})$
13:         **end for**
14:         Update weights $\theta \leftarrow \theta + \eta \cdot \Delta$, reset $\Delta = 0$
15:         From time to time copy weights into target network $\theta_{target} \leftarrow \theta$
16:     **end if**
17:     Choose action $A_t \sim \pi_\theta(S_t)$
18: **end for**=0

---

array of (4,64,64) as for the modified observation space of the robot. Figure 3 shows the modified observation space.

Another observation is that the explored map, called the global observation, could greatly help navigate the robot for exploration. Firstly we define frontiers as the boundary between explored free space and the unexplored space, which means that the robot can only gain more information if it moves to the frontiers. The worst problem occurs when there aren't any frontiers shown in the local map and the robot couldn't decide which direction to go. To overcome this issue, the simplest strategy is that we add the global explored map into the robot's observation space as a global observation. To keep the same dimension, we first generate the frontiers, obstacles and the pose of the robot in the image and resize it as (64,64), then it's stacked into the observation space. In this case, the size of our observation space is (5,64,64), consisting of 4 last frames of local map and current frame of global explored map. Figure 4 illustrates the procedure we generate the global explored map.

### 4.3 Reward Design

#### 4.3.1 Information Gain

From section 3, our original reward is defined as information gain $I(a_t|s_t)$, where $I(a_t|s_t) = -H(s_{t+1}|a_t, s_t) + H(s_t)$. Since the transition probability is deterministic, $s_{t+1} = \delta(P(s_{t+1}|a_t, s_t) \equiv 1)$. $H(s_t)$ is defined as the entropy of the whole map, where $H(s_t) = -\Sigma_{(x,y)} p(x, y) log(p(x, y))$. And $p(x, y)$ is defined as

$$p(x, y) = \begin{cases} 1 & explored \\ 0.5 & unexplored \end{cases} \tag{5}$$

In this case, if an unexplored area is explored after action $a_t$, then the information gain will be proportional to the size of the area, otherwise the information entropy won't change. From this point of view, the reward function aims to decrease the information entropy of the whole map so as to determine the structure of the map. The final reward is simplified as the area of newly explored space since it's directly proportional to the information gain.

#### 4.3.2 Artificial Potential Field

Artificial Potential Field(APF) is a general approach in path planning. In our project, we introduce APF to serve as a external reward so as to navigate the robot when it gets stuck in local corner. APF consists of Attractive Field and Repulsive Field, which are generated by the goal and obstacle

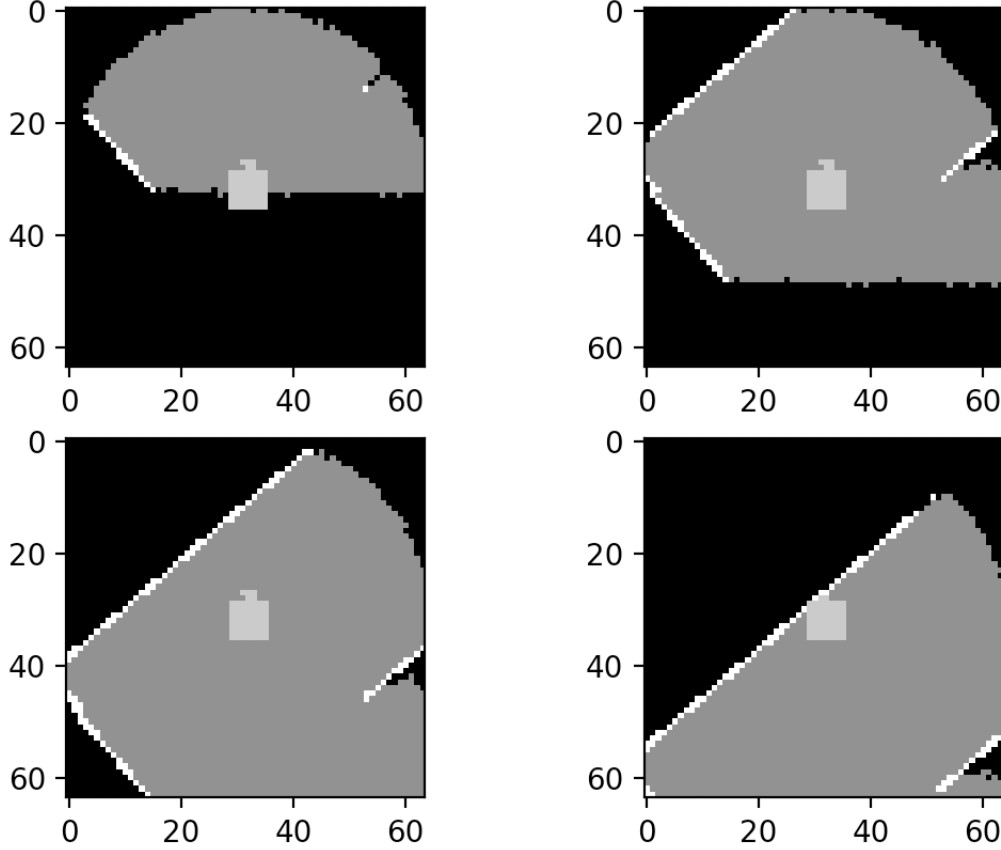

Figure 3: Sequence of Observation Space

correspondingly. Firstly, we denote $p_t$ as the pose of the robot at time $t$, and $E(p_t)$ as the energy at point $p_t$ in APF, where $E(p_t) = E_{att}(p_t) + E_{rep}(p_t)$. $E_{att}(p_t), E_{rep}(p_t)$ represent the energy at point $p_t$ in Attractive Field and Repulsive Field separately. And the formula of $E_{att}(p_t), E_{rep}(p_t)$ is as below.

$$E_{att} = \theta d^2(p_t, p_{goal}) \tag{6}$$

$$E_{rep} = \frac{\beta}{d^2(p_t, p_{obs}) + \xi} \tag{7}$$

where $d(p_1, p_2)$ means the distance between $p_1$ and $p_2$, $\theta, \beta$ are the hyperparameters determining the shape of the APM, and $\xi$ is a small number to avoid the denominator to be 0.

The original APF algorithm will let the agent move in the direction of the derivative of $E(p_t)$ so as to achieve the goal. Since the information gain will increase only if the robot move to the frontier, so we choose the center of the closest frontier as the goal. Figure 5 shows the procedure how we generate the Artificial Potential Field. To interpolate this idea into reward function, we proposed a novel approach to compute the reward function. Instead of computing the derivative, we compute the Energy difference between current frame and next frame as $\Delta(a_t, s_t) = E(p_t) - E(p_{t+1}|a_t)$, which means $\Delta(a_t, s_t)$ would be positive only if the energy decreases, so as to navigate the robot to the closest frontier. To make sure the reward is positive and normalize it, we compute our reward via softmax.

$$R_e(a_t, s_t) = \frac{exp(\Delta(a_t, s_t))}{\Sigma_{a_t} exp(\Delta(a_t, s_t))} \tag{8}$$

Then the final reward function is a combination of information gain and Artificial Potential Energy difference. Here $\lambda$ is a hyparameter to determine the weight of the external reward.

$$R(a_t, s_t) = I(a_t|s_t) + \lambda R_e(a_t, s_t) \tag{9}$$

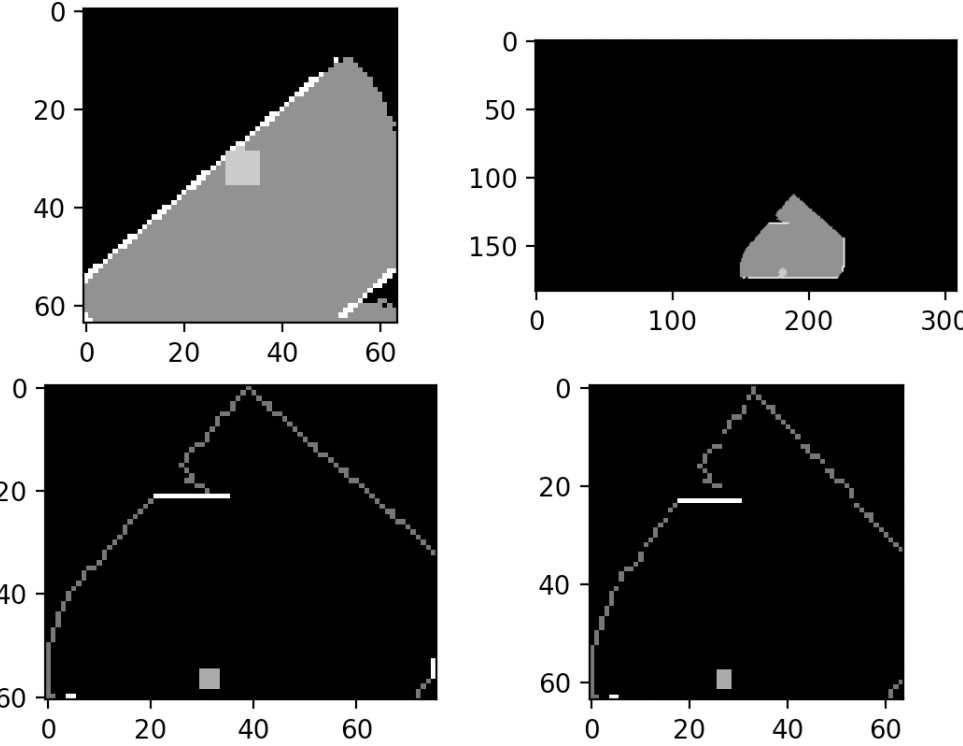

Figure 4: The generation procedure of global observation. (1) shows the local map. (2) shows the global explored map. (3) shows the processed frontiers,obstacles and pose of robot. (4) shows the final resized global observation.

## 5 Experiments

### 5.1 Experiment Setup

In this section, we report experimental results regarding the question: Whether our reinforcement learning algorithm can be helpful in accelerating the exploration process? We implement the autonomous exploration task to answer the question.

In our model, $s_t$ is $l \times l\ m^2$ rectangular area centred at the robot's position and its orientation is the same as robot's orientation, and $a_t$ contains three discrete bins as $[forward, left, right]$. The robot is equipped with a laser with a range of $l_l$ meters and a horizontal field of view $l_a$ degrees.

### 5.2 Autonomous Exploration

In this part, we demonstrate the efficiency of our proposed algorithm for autonomous exploration. Autonomous exploration refers to the process of searching for unknown areas. In our experiment, the robot is expected to discover as much area as possible to collect more information within a limited time. We split HouseExpo into training and testing set with $24588$ and $10538$ maps respectively. The robot only observes a local rectangular map around it. And its observation is rotation-invariant where the robot is always facing forward. The results is shown in Fig.6. Fig.6 plots the mean reward in evaluation time. We run our experiments with $Baseline$, $Baseline + Artificial Potential Field Reward$, and $Baseline + Global Observation$.Generally we can see that after training round 5M steps, the maximum average reward achieves around 160. Due to lack of GPU devices, we don't have enough time to run over all the experiments. Generally we can see that our baseline can achieve good results and shows the efficiency of RL algorithms, while the APF reward not works quite well. More importantly, we can see that the evaluation results suffer from a high variance.

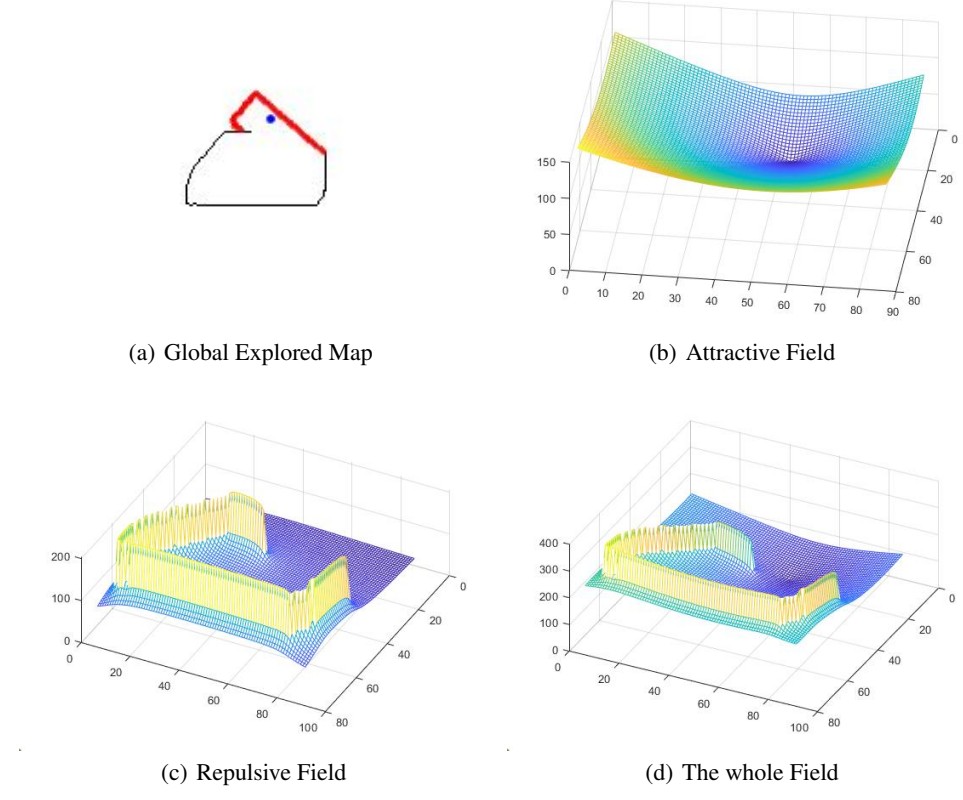

(a) Global Explored Map

(b) Attractive Field

(c) Repulsive Field

(d) The whole Field

Figure 5: Artificial Potential Field. (a) The global explored map, where the red area is the closest frontier we choose as the goal and the circle is the center of the goal. (b) Attractive Field generated by the goal. (c) Repulsive Field generated by the obstacle. (d) The whole artificial potential field.

## 6 Conclusion and Future Work

In conclusion, our proposed reinforcement learning algorithm illustrates promising prospect in solving autonomous exploration tasks. Generally, we explore two directions to improve our RL algorithm, encode history observation and design external reward to guide the policy. To encode the history observation, we concatenate the global explored map into our observation space, which may lead the robot to see existing frontiers that not exists in local map. And the external reward is designed as a guided policy search method to navigate the robot. The reason why these two methods don't improve the algorithm a lot may lie in lack of time to finetune the hyperparameters and the overfitting problem. During our experiments, we find that some RL algorithms are sensitive to hyperparameters and the performance may occur great variance. Generally, we think that more stable, explainable and sample efficient RL algorithm should be proposed.

### 6.1 Future work

The definition of Probability map in this paper is that the probability of unexplored is 0.5 while probability of explored area is 0 or 1. Calculated by the binary entropy function, the Shannon entropy of the unexplored area is 1, and the entropy of the explored area is 0. At this time, the formula of information gain was degenerated to the sum of unexplored pixel. The effectiveness of uncertainty estimating of the theorem is decreased. Actually, in the information compute method based on frontier, the map is classified into two class manually. During the future work, we can utilize neural network to minimize the relative entropy between the probability density function of the current map and the future map under the premise of the previous search. In this case, the probability of unexplored area is continuously 0 to 1, not constantly 0.5.

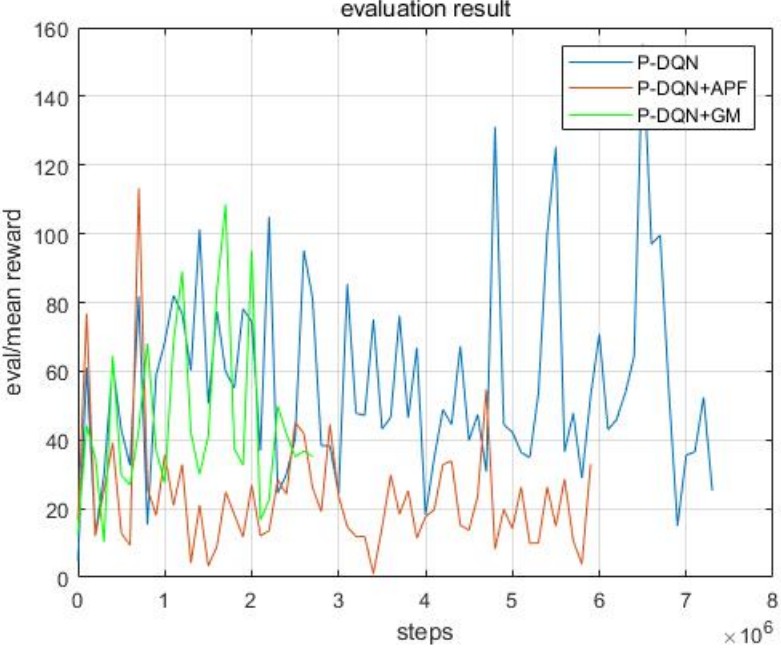

Figure 6: Evaluation Results

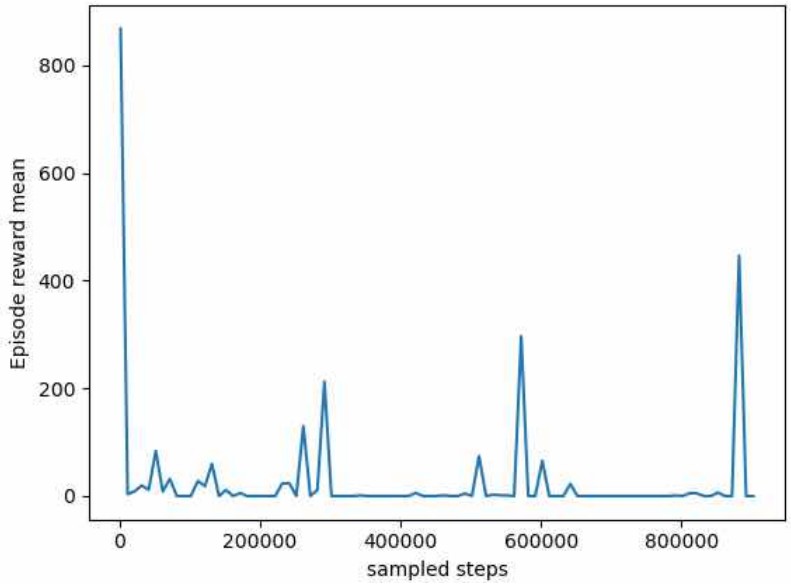

Figure 7: Training without reset

Besides, We did a simple experiment to what will happen training in this environment with out reset function. As shown in the figure 7., the agent has been in states without any reward for a long period for the agent has fully explored a sub-area and any actions will be no reward in this case. At the same time, the current policy with stochastic strategy are difficult to deviate from the current area, which makes the agent rotates in a area at most of the time. We believes that the curiosity mechanism can solve this problem. It uses the distilling mechanism to remove the dependence of the policy on the previous state. When the policy continuously produces bad actions, the algorithm will actively get rid of this state.

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
