# OpenReview forum: "Mobile Robots Autonomous Exploration with Reinforcement Learning"
_CUHK.edu.hk/2021/Course/IERG5350_

### Official Review · AnonReviewer3 · 2020-12-15
**Strong accept as the ideas are innovative and interesting while implementation can be improved**

**Rating:** 9
**Confidence:** 5

**Review:**

In this project, the authors implement a reinforcement learning framework to address the autonomous exploration task for mobile robots. Using the Double DQN with proportional prioritization as the baseline method, the authors explore two directions to improve the RL algorithm, by 1) encoding the global observation into the historical local observation, and 2) designing an Artificial Potential Field based reward to guide the policy. The proposed methods are innovative and interesting, although the experimental results show that the baseline has the best performance. I think this may be due to some issues in the implementation, which will be discussed below. In general, the ideas are innovative and interesting, while the implementation needs to be modified under careful consideration.

Here are some comments and suggestions for the authors:

1.In Section 4.2, some good points are illustrated about the reason you use the global observation for the robot exploration task. An original and interesting idea is proposed to combine global and local information together for a better exploration policy. I have some suggestions for the improvement of the implementation, though. In the current implementation, the global image is processed to extract the frontiers, obstacles and the robot and resized to (64,64), then it is stacked with the history frames as the observation. I think it doesn't make sense to stack the processed global information with the 4 stacked local observation, as the images have very different meanings. For example, the black region in the 4 stacked local frames show the unexplored area, while the black region in the processed global image is the area that doesn’t include frontiers, obstacles or the robot. You may think of other methods to encode the global observation, such as concatenate the image features extracted separately, instead of simply stacking together those images.

2.In Section 4.3.2, you provide a good idea to incorporate the traditional Artificial Potential Field (APF) method for path planning problem in your RL framework. It is impressive to incorporate the knowledge in the traditional path planning into the reinforcement learning process by customizing the reward function. I have some questions regarding your setting of goal position. You choose the center of the closest frontier as the goal in the APF based method. How do you define the “center” of the boundary? Is it the average (x, y) among all the points on the boundary? If so, there might be some problems in practice. Suppose the agent is surrounded by a frontier on which all the points have the same distance from the agent, then the center of the boundary would be right the position where the agent stands. In this situation, the attractive field will hint the movement of the agent and may result in undesired noises in your reward function. The authors are recommended to re-think the modification of the reward function for a better implementation of this idea.

3.The experimental results show that the baseline method outperforms the proposed improved versions and lacks a detailed explanation and description. For example, you may add the learning curves of your methods, and present some quantitative results of evaluation to better compare different methods, such as mean and variance of the returns in a number of episodes during evaluation. Also, you need to explain why you show the learning curve of training without resetting the environment in the end of the paper, which seems not a common practice in the field.

4.The paper is clear and organized in general. However, the introduction of related work needs to be more specific to make people outside this field quickly get familiar with the topic and understand your motivation. For example, you mention the Learned Map Prediction Based Robot Exploration, but do not give an explanation about what is “learned map” and how it is used.

5.The writing of the paper needs to be improved. There exist some typos and grammartical errors in the paper. For example, in the abstract, you mistakenly write “with a smart behaviour”. In the introduction part, you use “Due to the environments is different”, which should be “Due to the fact that the environment is different”. In the sentence “when facing on complex obstacles”, the use of the preposition is incorrect. In another sentence “The algorithms planning the trajectories while obtaining a better target position.”, the tense of the verb is incorrect.

6.The video lasts about 10 minutes and covers a lot of contents. You are recommended to make the content more concentrated and hence the video can be more concise (5 min as required).

In summary, my opinions about this work is concluded in terms of these four aspects:

- **Orininality: 10** (The ideas illustrated in this work are original and innovative, which could provide inspirations for researchers working in this direction. It is impressive to try combining knowledge in traditional path planning with reinforcement learning. )

- **Quality: 9** (The authors have made reasonable analysis of their improvements to the baseline for the robotic indoor exploration task. The reviewer believes that the implementation could be improved to achieve better experimental results.)

- **Clarity: 8** (The paper clearly defines the problem and introduces the methods. The experiment part can be improved by including more details.)

- **Significance: 8** (Autonomous indoor exploration is a hot topic in robotic research field. Other researchers in this field may draw inspiration from this work. Also, the methods could be potentially implemented on real mobile robots in the future.)

I think the work done in this project is overall excellent and satisfies the requirement of the course.

P.S.
Comments to AnonReviewer1: This review content was first submitted at 2020-12-15 15:15 and I have kept the record. Please do not copy and paste others’ work. Thanks!

---

### Official Review · AnonReviewer1 · 2020-12-15
**Successful implementation of RL in navigation task; Two innovative idea for improvements; Insufficient experiments; No good-enough performance**

**Rating:** 8
**Confidence:** 4

**Review:**

In this project, the authors implement a reinforcement learning framework to address the robot autonomous exploration task. The major contribution is that the authors explores to incorporate the history observation, global explored map and design of an external reward via Artificial Potential Field based on a Double DQN framework with proportional prioritisation.

There are some problems:
1) In section 4.2, to the reviewer's comprehension, the global observation which is actually the defined map should already contains information indicated in the last frames of local observation and may can also help the robot to find the path when getting stuck in the corner. And in authors' implementation, the resized global observation with processed features like frontiers is simply stacked with 4 last frames of the local map. The reviewer doubts that if the feature maps with different setting and meaning for pixels stacked would help. Meanwhile, the motivation to use history frames of local map and global map is illustrated but the reason of stacking them together is actually a little confusing for reviewers who lacks domain knowledge.
2) The idea of introducing Artificial Potential Field to reward design is innovative to reviewer's knowledge.
3) For the experiments, three algorithms - namely baseline, baseline with APF, baseline with GM - are not run for similar time steps so the comparison is not objective. Though author states that they have no enough time to run all the experiments due to the lack of device and time, it would be better to conduct an additional set of experiment with baseline + APF + GM.
4) From the results figure, it seems that the baseline shows best performance, and there is no strong evidence for the statement of promising prospect of APF and GM.
5) The authors should pay more attention to paper writing as there are some grammar mistakes, format error and even little typos throughout the paper. For example, the first sentence in the "Abstract" is actually not a sentence with a pretty long apposition but lacks predicate and object. Space signs are missed in Subsection of "Autonomous Exploration". Also, the figures are better to be placed in or near the corresponding section to make the paper easier to read and follow.
6) The video is a little long as the required length is 5 minutes. And it seems not a good idea to introduce oneself during the presentation as it may make the idea conveying inconsistent.

Overall, the idea and implementation satisfies the requirement of this course very well in reviewer's opinion. The authors successfully implemented DQN framework and proposes two innovative idea for further improvements though the preliminary experiment results showed no satisfying performance.

---

### Official Review · AnonReviewer2 · 2020-12-16
**good work with innovation, but need improvements in details**

**Rating:** 9
**Confidence:** 3

**Review:**

General:
The work try to design an efficient robot autonomous exploration algorithm based on reinforcement learning. This task is based on HouseExpo dataset, the data of which is collected from indoor areas. The author improve the Double DQN algorithm by adding more information and design external reward, so that the robot can efficiently explore uncertain environments with a smart behaviour.

Pros: 1. The idea of this task is quite interesting, and authors have innovation in the method, which is better than some other papers. 2. This paper is well structured, and the author has clearly plan about future work. 3. The method and the idea of the author are well organized in this paper, although some explanation of basic knowledge is ignored.

Cons: 1. Insufficient description and analysis on experimental results. 2. Need more attention to some details, for example, the overtime video, and the grammatical errors.